# Potential Mechanisms for Organoprotective Effects of Exogenous Nitric Oxide in an Experimental Study

**DOI:** 10.3390/biomedicines12040719

**Published:** 2024-03-23

**Authors:** Nikolay O. Kamenshchikov, Mariia L. Diakova, Yuri K. Podoksenov, Elena A. Churilina, Tatiana Yu. Rebrova, Shamil D. Akhmedov, Leonid N. Maslov, Alexander V. Mukhomedzyanov, Elena B. Kim, Ekaterina S. Tokareva, Igor V. Kravchenko, Alexander M. Boiko, Maxim S. Kozulin, Boris N. Kozlov

**Affiliations:** Cardiology Research Institute, Tomsk National Research Medical Center, Russian Academy of Sciences, Tomsk 634012, Russia; nikolajkamenof@mail.ru (N.O.K.); uk@cardio-tomsk.ru (Y.K.P.); lena.semenova.96@inbox.ru (E.A.C.); rebrova@cardio-tomsk.ru (T.Y.R.); shamil@cardio-tomsk.ru (S.D.A.); maslov@cardio-tomsk.ru (L.N.M.); sasha_m91@mail.ru (A.V.M.); ekim@cardio-tomsk.ru (E.B.K.); tokareva.kaierina@yandex.ru (E.S.T.); kravchenko27.10.94@gmail.com (I.V.K.); boiko.cardio@yandex.ru (A.M.B.); kozulin.max98@yandex.ru (M.S.K.); bnkozlov@yandex.ru (B.N.K.)

**Keywords:** nitric oxide, cardiac surgery, circulatory arrest, organ protection

## Abstract

Performing cardiac surgery under cardiopulmonary bypass (CPB) and circulatory arrest (CA) provokes the development of complications caused by tissue metabolism, microcirculatory disorders, and endogenous nitric oxide (NO) deficiency. This study aimed to investigate the potential mechanisms for systemic organoprotective effects of exogenous NO during CPB and CA based on the assessment of dynamic changes in glycocalyx degradation markers, deformation properties of erythrocytes, and tissue metabolism in the experiment. A single-center prospective randomized controlled study was conducted on sheep, *n* = 24, comprising four groups of six in each. In two groups, NO was delivered at a dose of 80 ppm during CPB (“CPB + NO” group) or CPB and CA (“CPB + CA + NO”). In the “CPB” and “CPB + CA” groups, NO supply was not carried out. NO therapy prevented the deterioration of erythrocyte deformability. It was associated with improved tissue metabolism, lower lactate levels, and higher ATP levels in myocardial and lung tissues. The degree of glycocalyx degradation and endothelial dysfunction, assessed by the concentration of heparan sulfate proteoglycan and asymmetric dimethylarginine, did not change when exogenous NO was supplied. Intraoperative delivery of NO provides systemic organoprotection, which results in reducing the damaging effects of CPB on erythrocyte deformability and maintaining normal functioning of tissue metabolism.

## 1. Introduction

The challenges in cardiac surgery and the comorbidity of patients are steadily increasing every year, which requires the development of new methods for organ protection to prevent postoperative complications and improve the outcomes of interventions. Cardiac surgery in most cases requires cardiopulmonary bypass (CPB), and in some cases, surgery on the aortic arch and hypothermic non-perfusion circulatory arrest (CA). Extracorporeal perfusion has a complex negative effect on the microvasculature as a result of ischemia–reperfusion injury and oxidative stress, leading to the development of organ injury [1,2], which may be based on a deficiency of endogenous nitric oxide (NO) developed as a result of the glycocalyx and endothelium degradation.

Glycocalyx degradation is currently considered the cornerstone of endothelial dysfunction, triggering the process of local microcirculation disorder with subsequent organ injury due to vasoconstriction, leukocyte adhesion, activation of the immune response, and compartmentalization of inflammation in tissues [3]. The glycocalyx contains various chemokines, receptors, growth factors, and enzymes, including endothelial nitric oxide synthase (eNOS) [3]. For the proper functioning of eNOS, an intact endothelial glycocalyx is required, and NO production is completely blocked when its components are removed [4]. In addition to the direct damaging effect on the cellular and supracellular components of the vascular wall, CPB promotes the development of hemolysis. Hemolysis, in turn, causes an increase in the concentration of free hemoglobin in plasma, which negatively affects blood rheology and acid-base balance, and also reduces the level of intravascular NO due to binding to free hemoglobin [1,5,6].

Erythrocyte deformability plays a key role in the development of hemolysis and in maintaining normal interactions between red blood cells and tissues.

Erythrocyte deformability is impaired under the influence of mechanical factors, hypothermia, and oxidative stress associated with CPB [1,7,8]. Experimental studies showed that NO is a critical factor in determining the mechanical behavior of erythrocytes and helps normalize the deformability of erythrocytes under oxidative stress [6,7,9].

Thus, CPB provokes a violation of the synthesis and metabolism of NO with the development of its deficiency in the microcirculatory and systemic beds, leading to the development of organ injury. It seems promising to study the possibility of using exogenous NO to replenish its deficiency under glycocalyx degradation and optimize the deformability of erythrocytes to improve transcapillary substrate exchange in an experiment with CPB and CA modeling.

The aim of our study was to investigate the potential mechanisms of the systemic organoprotective effects of exogenous NO during CPB and CA based on the assessment of dynamic changes in glycocalyx degradation markers, the deformability of erythrocytes, and tissue metabolism in the experiment.

## 2. Materials and Methods

A single-center prospective randomized controlled study was conducted to identify the probable mechanisms of the organoprotective effects of exogenous NO in modeling operations with CPB, CPB, and CA (CPB + CA). The work was carried out on the basis of the laboratory of critical care medicine, the department of cardiovascular surgery, and the department of anesthesiology and critical care of the Federal State Budgetary Scientific Institution “Tomsk National Research Medical Center of the Russian Academy of Sciences”, Cardiology Research Institute. This study was approved by the Ethics Committee of the Cardiology Research Institute, Tomsk NRMC, Protocol No. 230, dated 28 June 2022.

The object of this study was Altai breed sheep weighing 30–34 kg, *n* = 24. Before the start of the experiment, all animals were kept in a conventional vivarium. All procedures were performed in a testing laboratory according to [10].

The study methodology was developed in accordance with international guidelines for randomized controlled trials. The sequence of events and study methodology are presented below.

Recruitment of animals for this study;Examination by a veterinarian and admission to participate in this study;Randomization;Induction of anesthesia and the onset of mechanical ventilation;NO (nitric oxide) conditioning or no NO supply during the entire main phase of the experiment;Anesthesia;Conducting CPB or CPB + CA;Tracking of clinical events and characteristics; serial perioperative measurements of the biochemical variables;Collection of biopsy samples;Conservation of biopsy samples;Withdrawal of animals from the experiment;Laboratory study of the obtained biopsy samples.

Randomization was carried out by the method of sealed opaque envelopes. The number of envelopes was equal to the estimated sample size, *n* = 24. The processing distribution was prepared by an independent operator (a researcher not involved in this study) and hidden in opaque sealed envelopes that were consecutively numbered. Each envelope contained one code name: “CPB”, “CPB + NO”, “CPB + CA”, or “CPB + CA + NO”. On the morning of the experiment, when an envelope was opened, its contents were not disclosed. Thus, the animals were assigned to the group of perioperative NO conditioning with CPB or CPB + CA or to the group of the standard protocol CPB or CPB + CA. All animals were randomized into 4 equal groups of 6 sheep in each group.

CPB group: the standard protocol of mechanical ventilation and CPB was carried out, and CPB time was 90 min;CPB + NO group: NO was delivered immediately after tracheal intubation through the circuit of the ventilator at a dose of 80 ppm, and then at the start of CPB, NO was delivered to the extracorporeal circulation circuit at a dose of 80 ppm throughout the entire period of CPB (90 min), and after weaning from CPB, NO supply continued through the circuit of the ventilator at a dose of 80 ppm for 1 h;CPB + CA group: the standard protocol adopted in the clinic for mechanical ventilation, CPB, and hypothermic circulatory arrest was carried out; CPB time was 90 min, and there was 15 min of hypothermic circulatory arrest at 30 °C;CPB + CA + NO group: NO was delivered immediately after tracheal intubation through the ventilator circuit at a dose of 80 ppm, and then, at the start of CPB, NO was delivered to the extracorporeal circulation circuit for 90 min; CA was carried out with hypothermia. After reaching the target esophageal temperature of 30 °C, the descending aorta was occluded and, thus, non-perfusion circulatory arrest was simulated for 15 min. The perfusion index was then reduced to 1 L/min/m^2^. When completing CA, NO delivery to the extracorporeal circuit at a dose of 80 ppm was resumed and maintained until normothermia was achieved (CPB + CA duration was 90 min); and after weaning from CPB, NO was supplied again through the ventilator circuit at a dose of 80 ppm for 1 h.

### 2.1. Method of Anesthesia and Cardiopulmonary Bypass

The experiment began with the sevoflurane mask induction of anesthesia in all groups. After reaching the target level of anesthesia, preoperative preparation of shaving and processing of the surgical area was performed. To carry out the induction of anesthesia with the aseptic technique, catheterization of the saphenous vein of the hind limb with an 18 G catheter was performed. General anesthesia was induced by fractional administration of propofol 1% at a dose of 5 mg/kg. While maintaining spontaneous breathing, direct laryngoscopy followed by orotracheal intubation with an endotracheal tube 6.5 mm with an introducer was performed. The endotracheal tube was fixed. Then, mechanical ventilation was performed with a 760 ventilator (Puritan Bennett, KY, USA).

In the CPB + NO and CPB + CA + NO groups, immediately after tracheal intubation, the delivery of nitric oxide was initiated through a modified breathing circuit at a dose of 80 ppm. Throughout the experiment, in order to maintain anesthesia, a continuous intravenous infusion of propofol 1% at a dose of 5 mg/kg/h was performed. Neuromuscular blockade was achieved with pipecuronium bromide at a dose of 0.1 mg/kg. Throughout the experiment, extensive monitoring during anesthesia was used, including electrocardiogram monitoring, invasive blood pressure monitoring, pulse oximetry, continuous monitoring of end-tidal carbon dioxide (etCO_2_), and thermometry using a BSM-4104A bedside patient monitor (Nihon Kohden, Tokyo, Japan); the urine flow rate was also taken into account. The temperature sensor was placed in the esophagus. The common carotid artery was surgically harvested and catheterized with a 20F catheter. The internal jugular vein underwent cannulation with a double-lumen 7F catheter. Surgical access was performed with right 4th–5th intercostal space thoracotomy.

A cardiopulmonary bypass was performed using anHL20 CPB machine (Maquet, Hechingen, Germany). The CPB machine was connected according to the “aorta—superior vena cava—inferior vena cava” scheme. Mean arterial pressure during CPB was maintained at 50–60 mm Hg.

In groups of animals that did not simulate CA, CPB was performed under normothermic conditions, and esophageal temperature was maintained at a level of 36–36.6 °C. In two other groups of animals with simulated CA, CPB was carried out with hypothermia. After reaching the target esophageal temperature of 30 °C, the descending aorta was occluded and, thus, non-perfusion CA was simulated for 15 min. Next, the descending aorta was unclamped, and then reperfusion and warming to 36.6 °C were carried out. The cumulative duration of CPB in all groups was 90 min. To ensure hypocoagulability during CPB, heparin at a dose of 300 U/kg was used, maintaining the activated clotting time > 450 s.

### 2.2. Nitric Oxide Conditioning

For nitric oxide delivery, a special device for plasma–chemical synthesis of nitric oxide was used. To ensure safety, NO/NO_2_ concentrations, as well as the level of methemoglobin (MetHb) in the blood, were continuously monitored throughout the entire experiment. An increase in NO_2_ to 2 ppm or more was considered critical; when the mentioned concentration was reached, the delivery of NO was supposed to be discontinued.

In the CPB + NO group, NO was delivered immediately after tracheal intubation through the circuit of the ventilator at a dose of 80 ppm, and then at the initiation of CPB, NO was delivered to the extracorporeal circulation circuit at a dose of 80 ppm throughout the entire period of CPB (90 min), and after the restoration of blood flow at the reperfusion stage, NO supply continued through the circuit of the ventilator at a dose of 80 ppm for 1 h (Figure 1).

In the CPB + CA + NO group, NO was delivered immediately after tracheal intubation through the ventilator circuit at a dose of 80 ppm throughout the entire experiment and then, at the start of CPB, NO was delivered to the extracorporeal circulation circuit for 90 min until hypothermic CA was initiated. During CA, when the target body temperature, an esophageal temperature of 30 °C, was reached, the perfusion index decreased to 1 L/min/m^2^, and the descending aorta was occluded for 15 min, and NO delivery was not performed. When completing CA, NO delivery to the extracorporeal circuit at a dose of 80 ppm was resumed and maintained until normothermia was achieved (the cumulative duration of CPB and CA was 90 min). After weaning from CPB, NO was supplied again through the ventilator circuit at a dose of 80 ppm for 1 h.

NO was delivered through the ventilator circuit into the inhale tube consisting of two circuit tubes connected by an absorber, which was filled with soda lime. The first tube was joined to the absorber with an angled Luer connector. NO was supplied to this connector. Next, the second tube was attached to the absorber. The other end of the second tube was connected to a Y- adapter with a straight Luer connector, from where gas was taken for continuous monitoring of NO and NO_2_ levels. The inhale and exhale tubes were connected with a Y- adapter. The inhale tube was equipped with a hydrophobic virus–bacterial filter with a Luer connector (Figure 1A).

NO was delivered to the CPB machine through the gas–air line. A gas flow regulating line consisting of two parts connected by two ¼ Luer-lock adapters was attached to the gas blender. The NO supply line was joined to the Luer connector located closer to the gas blender, and the gas sample intake line was connected to the Luer connector located closer to the oxygenator. The other end of the gas flow regulating line was connected to the oxygenator. In the oxygenator, venous blood was saturated with oxygen and NO. Next, the obtained arterial blood was supplied through the arterial line to the aorta (Figure 1B).

Collecting blood samples for biochemical analysis was carried out at the following stages: 1—before the initiation of CPB; 2—at the initiation of CPB; 3—after weaning from CPB.

To measure the glycocalyx degradation marker, heparan sulfate proteoglycan (HSPG), venous blood was collected in BD Vakuteiner tubes (BD, New Jersey, USA), and then centrifugation was performed for 10 min at 3500 rpm. Next, an enzyme-linked immunosorbent assay (ELISA) was performed to quantitatively measure HSPG in vitro in plasma using an ELISA kit (Cloud-Clone Corp, Houston, TX, USA).

To measure the endothelial dysfunction marker, asymmetric dimethylarginine (ADMA), venous blood was collected in BD Vakuteiner (BD, Bergen, NJ, USA) tubes and then centrifuged for 10 min at 3500 rpm. Next, the serum was frozen and stored at −20 °C. Quantitative determination of ADMA was carried out with ELISA using an ADMA Xpress ELISA KR7860 (Immundiagnostik AG, Bensheim, Germany). This assay is based on the competitive enzyme immunoassay technique.

To study erythrocyte deformability, the coefficient of microviscosity and polarity in the areas of lipid–lipid (CMLLI; CPLLI) and protein–lipid interactions (CMPLI; CPPLI) were assessed. To determine the coefficient of microviscosity and polarity of erythrocyte membranes, venous blood was collected immediately after intubation before the initiation of CPB and after weaning from CPB. Blood was collected into vacutainer tubes containing lithium heparin (17 IU/mL) sprayed on the walls. Blood samples were centrifuged at 1500 rpm for 10 min. After removing the plasma, erythrocytes were washed 3 times with cooled saline; each time the erythrocytes were sedimented at 1500 rpm for 10 min. Erythrocyte membranes were obtained by hypoosmotic hemolysis. The amount of total protein in the suspension of erythrocyte shadows was determined with the Micro–Lowry method and Ohnishi S.T. modification using Sigma-Aldrich reagents (Sigma-Aldrich, St. Louis, MO, USA). To assess spectral characteristics, samples of erythrocyte membranes were diluted in 10 mM Tris-HCl buffer (pH = 7.4) to a final protein concentration of 0.3 mg/mL.

To study the structural properties of the lipid phase of erythrocyte membranes, spectral characteristics of the interactions between membrane and pyrene fluorescent probes (Sigma-Aldrich, St. Louis, MO, USA) were assessed on a Cary Eclipse fluorescence spectrometer (Varian, Inc., Palo Alto, CA, USA). A total of 20 μL of a 10 μM alcohol solution of the pyrene probe was added to 2 mL of erythrocyte membrane suspension. Membrane microviscosity of annular and total lipids was assessed by the degree of pyrene excimerization calculating the excimer-to-monomer fluorescence intensity ratio (J470/J370) at an excitation wavelength (λ_Ex_) of 285 and 340 nm, respectively. Polarity was analyzed by the excimer-to-monomer vibration peak amplitude ratio (J390/J370) at excitation wavelengths (λ_Ex_) of 285 and 340 nm, respectively.

To measure the adenosine triphosphate (ATP) concentration in the heart and lung tissues, biopsy samples were taken 1 h after the completion of CPB and the restoration of spontaneous circulation. The biopsies were then frozen in liquid nitrogen. The obtained samples were homogenized in liquid nitrogen and centrifuged for 10 min at 3000 rpm and 2 °C. Next, a supernatant liquid was obtained, which was collected and neutralized, and the sample volume was adjusted to 2 mL. ATP measurement was carried out using the ATP Bioluminescent Assay Kit (Sigma-Aldrich, St. Louis, MO, USA) on a Lucy-2 luminometer (Anthos Labtec Instruments GmbH, Salzburg, Austria).

The lactate concentration measurement was carried out by enzyme immunoassay using the L-Lactate Assay Kit (Sigma-Aldrich, St. Louis, MO, USA) on a multifunctional microplate reader Infinite 200 (Tecan Austria GmbH, Grödig, Salzburg, Austria). The principle of this assay is based on lactate oxidation, catalyzed by lactate dehydrogenase. Nicotinamide adenine dinucleotide (NAD) + hydrogen (NADH) formed during this reaction reduced a formazan reagent, and the color intensity of the resulting solution was proportional to the lactate concentration in the sample.

The normal distribution of quantitative variables was examined using the Shapiro–Wilk test. If the variables had a normal distribution, they were described by the mean value and standard deviation, mean ± SD; otherwise, by the median (Me) and interquartile range median (Q1; Q3). Significant differences in quantitative variables for independent and dependent samples were analyzed using Student’s *t*-test in case of the normal distribution of the variable in all compared groups, or otherwise using the Mann–Whitney U test for independent samples and the Wilcoxon signed-rank test for dependent samples. The significance threshold for testing hypotheses was *p* = 0.05.

In this study, statistical intragroup and intergroup analysis was performed. The “CPB” and “CPB + NO” groups, as well as the “CPB + CA” and “CPB + CA + NO” groups, were compared, which made it possible to assess the effect of NO on organs and systems when performing different models of mechanical perfusion.

## 3. Results

The following variables were analyzed: the CMLLI and CMPLI and the CPLLI and CPPLI of erythrocyte membranes in groups before the initiation of CPB (the start of the experiment) and after weaning from CPB. The data are presented in Table 1 and Figure 2.

At the beginning of the experiment, the coefficients of the microviscosity and polarity of erythrocyte membranes were comparable in all groups.

When analyzing CMLLI within the groups and between the “CPB” and “CPB + NO” groups before and after CPB, no differences were revealed. Analysis of CMPLI in the “CPB” group revealed a decrease after weaning from CPB by 55% (*p* = 0.009). In the “CPB + NO” group, no changes in CMPLI were observed at different stages of the experiment.

When analyzing CPLLI between the “CPB” and “CPB + NO” groups, no intergroup differences were revealed at different stages of the experiment. In the “CPB” group, there was a decrease in CPPLI after weaning from CPB by 32% (*p* = 0.013) in comparison with the stage before the initiation of CPB. In the “CPB + NO” group, there were no changes observed in CPPLI after weaning from CPB in comparison with the stage before CPB.

Analysis of the CMLLI in the “CPB + CA” and “CPB + CA + NO” groups did not show either changes at the observation stages of the experiment or intergroup differences. There was a 35% decrease in CMPLI in the “CPB + CA” group after weaning from CPB (*p* = 0.049). In the “CPB + CA + NO” group, no dynamics in CMPLI were detected after weaning from CPB in comparison with the stage before the initiation of CPB.

When analyzing the CPLLI of erythrocyte membranes in the “CPB + CA” and “CPB + CA + NO” groups, no significant differences were revealed at different stages within the groups or between the groups. In the area of protein–lipid interactions in the “CPB + CA” group, there was a significant decrease in the coefficient of polarity by 38% (*p* = 0.023) after weaning from CPB in comparison with the stage before CPB. In the “CPB + CA + NO” group, there were no changes in the CPPLI of erythrocyte membranes before CPB and after weaning from CPB.

### 3.1. HSPG as a Marker of Glycocalyx Degradation

The dynamics of HSPG levels in the blood were analyzed at different stages of the experiment: before CPB, at the initiation of CPB, and after weaning from CPB. The data are presented in Table 2.

At the beginning of the experiment, the levels of HSPG in the blood were comparable in the “CPB” and “CPB + NO” groups, as well as in the “CPB + CA” and “CPB + CA + NO” groups. There were no changes in the levels of HSPG at different stages in the groups with and without NO therapy, and no intergroup differences were revealed.

### 3.2. ADMA as a Marker of Endothelial Dysfunction

Analysis of ADMA levels in the “CPB”, “CPB + NO”, “CPB + CA”, and “CPB + CA + NO” groups at different stages of the experiment: before CPB, at the initiation of CPB, and after weaning from CPB did not show the dynamics of this variable. It did not reveal intergroup dynamic differences either (Table 3).

### 3.3. Tissue Concentrations of ATP and Lactate in Cardiac and Lung Biopsies

Analysis of ATP levels in tissues showed a 64% (*p* = 0.0019) higher concentration in cardiac tissues and 46% (*p* = 0.0008) higher concentration in lung tissues after weaning from CPB at the background of exogenous NO therapy in comparison with the generally accepted technique of CPB (Table 4). In the “CPB + CA” and “CPB + CA + NO” groups, the ATP concentrations in all studied tissues were comparable (Figure 3).

The lactate concentrations in heart tissues in the “CPB + NO” group were 29% (*p* = 0.0073) lower compared to the “CPB” group (Figure 4). The lactate concentrations in lung tissues in the “CPB + NO” group were slightly lower than in the “CPB” group, but there was no statistical difference (*p* = 0.059) (Figure 4).

The lactate concentrations in heart and lung tissues in the “CPB + CA” and “CPB + CA + NO” groups were comparable (Figure 4).

## 4. Discussion

Nitric oxide is one of the most important biological molecules in the human body. The NO molecule, due to its unique structure, has virtually no barriers in either cells or tissues, which makes it an ideal messenger of intercellular and intertissue interactions. NO is a universal transmitter that is involved in the development of physiological and pathological processes [11]. Under normal oxygenation, NO is generated from three types of NOS: endothelial (eNOS), neuronal (nNOS), and macrophage (mNOS) [12]. eNOS is present in vascular endothelium. When intracellular calcium increases, NO synthesis is immediately activated. Soluble guanylate cyclase is then activated, which, in turn, reduces intracellular calcium levels. A decrease in calcium levels leads to vascular smooth muscle relaxation. NO also has an inhibitory effect on platelet aggregation and inhibits the activation and adhesion of leukocytes and their release of pro-inflammatory agents. The complex action of NO released from endothelial cells improves blood flow in a specific area of microvasculature [13]. Thus, maintaining a normal level of NO in the vascular bed ensures the maintenance of an adequate state of microvasculature and, accordingly, tissue oxygenation and metabolism.

NO is one of the most important substances that interact with red blood cells [14,15]. Red blood cells act as an important interorgan communication network with various functions, including the control of systemic NO metabolism, redox regulation, and blood rheology [1,6]. The main source of NO is endothelial cells, but NO can also be produced by red blood cells themselves [16]. One of the effects of NO on erythrocytes is their protection from sub-hemolytic mechanical damage [17]. NO increases erythrocyte deformability [6] and also protects erythrocytes from eryptosis [18]. Presumably, this antieryptotic effect prevents further reduction in erythrocyte deformability.

Erythrocyte deformability is the ability of cells to adapt their shape to dynamically changing flow conditions to minimize their resistance to flow [19]. This property of red blood cells is important for their passage through capillaries. Reduced deformability (increased rigidity) leads to impaired perfusion and oxygen delivery to peripheral tissues [19,20,21,22]. Red blood cells perceive mechanical forces, leading to deformation, and respond to them by releasing vasoactive mediators, which also affect blood flow by influencing vascular endothelial or smooth muscle cells. For instance, ATP exported from deformed red blood cells affects endothelial cells through purinergic receptors and increases e-NOS activity. Thus, red blood cell deformation is an important stimulus for the export of ATP from red blood cells and allows them to dilate arterioles and easily cross capillaries, which, in some cases, are smaller in diameter than the red blood cells themselves [19].

The effect of NO on erythrocyte stiffness depends on NO concentration [23]. When NO concentration was 10^−7^ M, the deformability of erythrocytes improved; when it was 10^−5^ M, membrane lipid fluidity decreased, and when NO concentration was 10^−3^ M, an increase in methemoglobin concentration and a decrease in erythrocyte deformability was observed, although membrane fluidity and lipid peroxidation did not change compared to the control group. Thus, NO is the main physiological regulator of the rheological properties of blood due to its direct effect on erythrocyte deformability [14].

Deformation, mediator, and anti-adhesive effects of erythrocytes are disrupted in pathology in accordance with mechanisms that remain poorly understood [19].

Hypothermia, hemodilution, and mechanical stress during CPB and CA synergistically reduce the deformability of erythrocytes and can impair microcirculation and oxygen supply to tissues [8].

Our study made it possible to investigate the change in erythrocyte deformability under CPB and CA. The CMPLI, CMLLI, CPPLI, and CPLLI of erythrocyte membranes were analyzed. The results of our study demonstrate a significant decrease in CMPLI by 55% (*p* = 0.009) and CPPLI by 32% (*p* = 0.013) after weaning from CPB compared to the values before the initiation of CPB, which indirectly confirms the degradation of erythrocyte deformability under CPB. This fact is probably due to the deficiency of endogenous NO during CPB since exogenous NO supply made it possible to reduce the negative effect of CPB on erythrocyte deformability. In the “CPB + NO” group, the CPPLI, CPLLI, CMPLI, and CMLLI after weaning from CPB were comparable to the values before the initiation of CPB.

Similar dynamics of erythrocyte deformability can be observed in the “CPB + CA” and “CPB + CA + NO” groups. Thus, in the “CPB + CA” group, the CMPLI after weaning from CPB decreased by 35% (*p* = 0.049) in comparison with the stage before the initiation of CPB, and the CPPLI decreased by 38% (*p* = 0.023). During therapy with exogenous NO, the CMPLI and CPPLI after weaning from CPB were comparable to the initial variables. Thus, the supply of exogenous NO made it possible to reduce the negative effect of CPB and CA on the deformability of erythrocytes.

A patient’s circulating blood volume through the extracorporeal CPB circuit is associated with various kinds of pathophysiological disorders [24,25]. In particular, exposure to abnormal shear stress on the blood flow and interactions with artificial contact surfaces of the CPB circuit leads to coagulation and complement activation, pro-inflammatory activation, endothelial cell death [26], and platelet and leukocyte activation [27,28]. The systemic inflammatory response is activated and the subcellular structure, glycocalyx, is damaged. Glycocalyx damage is considered to be the initial stage of inflammation and leads to endothelial cell dysfunction. Vascular endothelial dysfunction exacerbates the inflammatory response, which further contributes to glycocalyx damage [29]. Glycocalyx degradation causes increased endothelial permeability, increased pro-inflammatory cell migration, and the disruption of mechanotransduction and eNOS activity [30], resulting in the development of endogenous NO deficiency, which aggravates the development of endothelial dysfunction. Heparan sulfate proteoglycans are the main structural component of endothelial glycocalyx and serve to regulate vascular permeability, microcirculatory tone, leukocyte and platelet adhesion, and hemostasis [31]. ADMA is an endogenous amino acid and an analog of L-arginine that can inhibit e-NOS, causing a disruption of NO formation mechanisms in blood plasma and tissues. Impaired synthesis and availability of NO due to decreased eNOS activity is one of the main causes of endothelial dysfunction. Thus, an increased level of ADMA is a marker of endothelial dysfunction [32]. In our study, HSPG and ADMA levels did not change significantly after the completion of CPB and CA in all groups, which may be due to the relatively early collection of biological samples immediately after CPB completion. It cannot be excluded that sampling material later after surgery would have revealed differences in the levels of markers of glycocalyx degradation and endothelial dysfunction between groups against the background of NO supply.

Glycocalyx degradation, endothelial dysfunction, impaired erythrocyte deformability, and hemolysis as negative consequences of CPB and CA inevitably lead to the development of hypoxia and the disruption of tissue metabolism. A metabolic disorder, in turn, is manifested in a natural decrease in ATP concentration and an increase in lactate concentration. This is due to changes in metabolism under conditions of oxygen deficiency. The production of lactate as an energy source increases when the demand for oxygen and ATP exceeds its supply [33,34]. Changes in tissue metabolism with lactate accumulation and ATP decrease can be considered as one of the manifestations of ischemia–reperfusion syndrome during CPB. The developing ATP and lactate imbalance directly reflects the severity of tissue hypoxia, which has developed as a result of microcirculatory disorders, impaired erythrocyte deformability, endothelial dysfunction, and hemolysis during CPB and CA.

Exogenous NO supply during cardiac surgery in our study was associated with a decrease in the negative effect of CPB and CA on erythrocyte deformability. This effect, in addition to the direct effect on coefficients of microviscosity and polarity of erythrocyte membranes, was supposed to be confirmed when analyzing tissue metabolism as well. Our study demonstrated a higher ATP level in myocardial and lung tissues against the background of exogenous NO supplementation during CPB compared to the standard protocol for CPB. This corresponded to a lower lactate level in the heart tissue in the “CPB + NO” group compared to the “CPB” group. The data obtained indicate an improvement in tissue metabolism and a systemic organ protective effect mechanism resulting in exogenous NO supplementation.

Exogenous NO supply during CPB and CA was not associated with lower lactate concentration and higher ATP concentration in myocardial and lung tissues compared to the standard approach to CPB and CA management. The obtained results of lactate and ATP concentrations in myocardial and lung tissues during CA were probably due to a more pronounced inhibition of tissue metabolism with the combined effects of CPB and CA. The negative effect of such an additional factor as CA on tissue metabolism neutralizes the effect of exogenous NO on erythrocyte deformability confirmed in our study. The obtained data require the development of new protocols for CPB and CA management and the supply of exogenous NO for effective organ protection.

It is worth noting that this study was limited by the lack of data on molecular mechanisms of implementation of exogenous NO organoprotective effects.

## 5. Conclusions

The organoprotective effectiveness of exogenous NO was established when simulating cardiac surgery not only under CPB but also with hypothermic CA. The prevention of erythrocyte deformability deterioration is a potential mechanism for the organ protective effect of exogenous NO. NO supply during surgery with CPB is associated with less severe tissue metabolic disorders, as evidenced by lower lactate levels and higher ATP levels in tissues. Thus, the use of NO during surgery with CPB and CA is a promising direction in cardiac anesthesiology and is aimed at reducing the development of complications in the postoperative period. The obtained experimental results are the basis for further research in clinical practice.

## Figures and Tables

**Figure 1 biomedicines-12-00719-f001:**
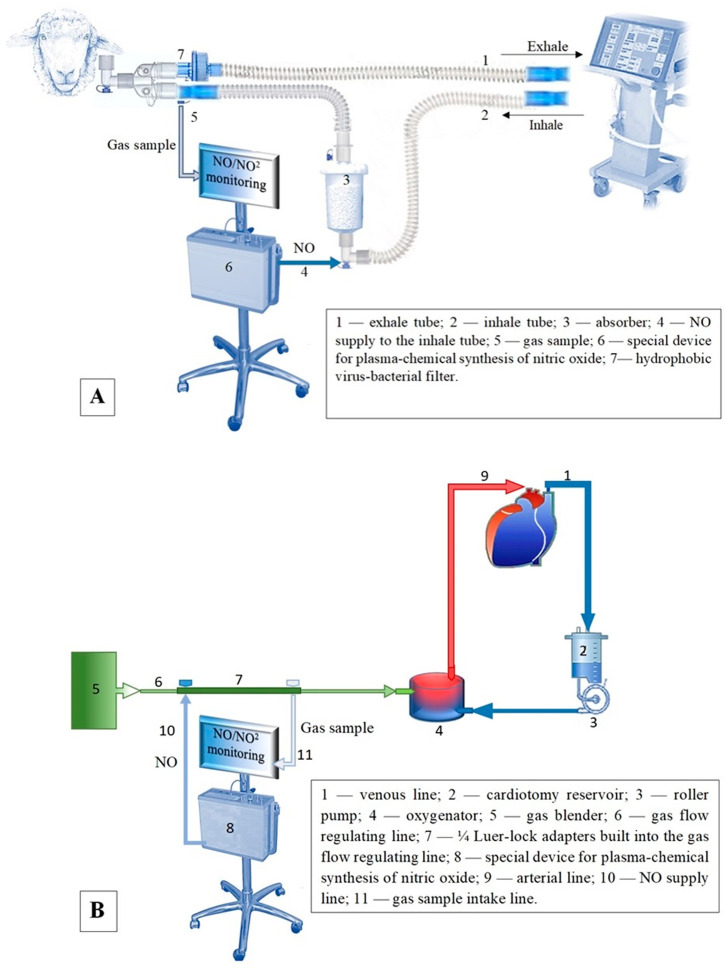
NO delivery: (**A**) through the ventilator circuit; (**B**) through the gas–air line of the CPB machine.

**Figure 2 biomedicines-12-00719-f002:**
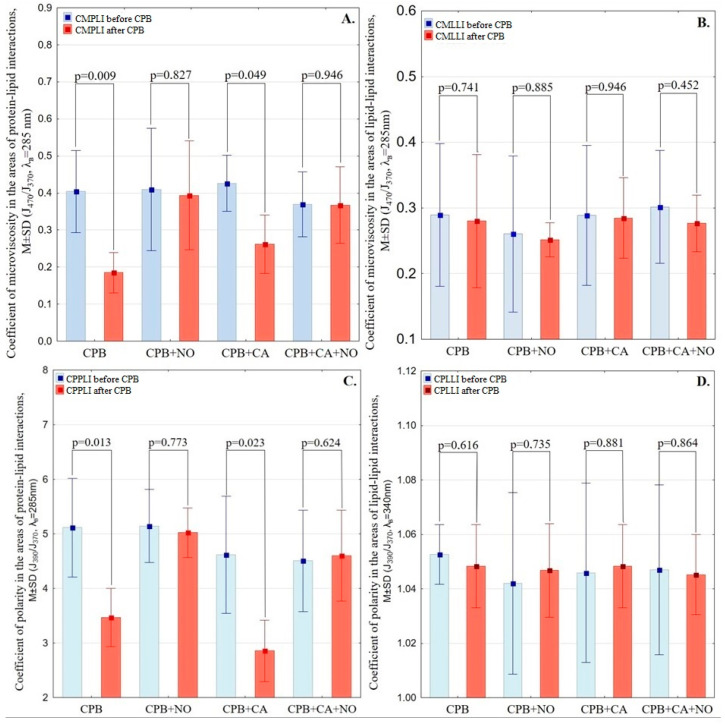
(**A**) coefficient of microviscosity in the areas of protein–lipid interactions of erythrocyte membranes at different stages of the experiment; (**B**) coefficient of microviscosity in the areas of lipid–lipid interactions of erythrocyte membranes at different stages of the experiment; (**C**) coefficient of polarity in the areas of protein–lipid interactions of erythrocyte membranes at different stages of the experiment; (**D**) coefficient of polarity in the areas of lipid–lipid interactions of erythrocyte membranes at different stages of the experiment.

**Figure 3 biomedicines-12-00719-f003:**
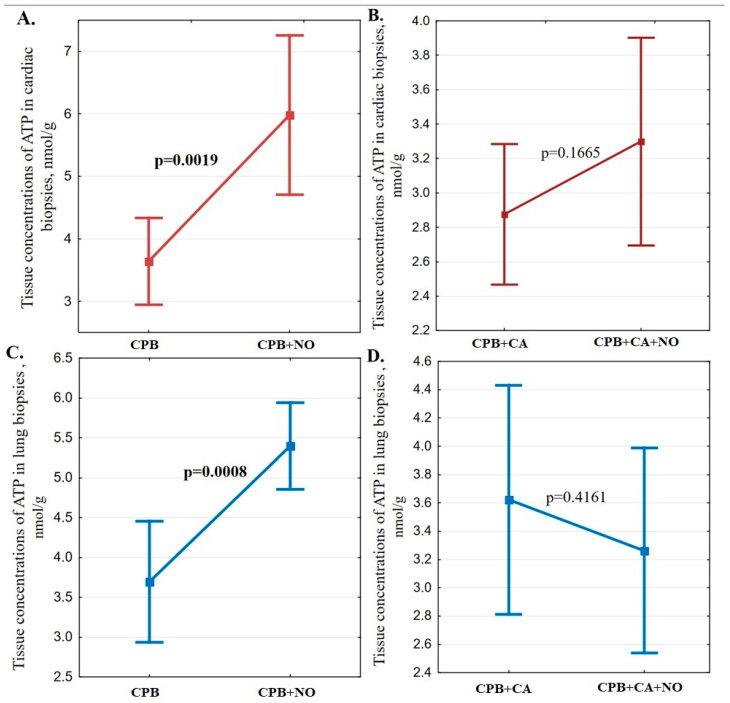
(**A**) A comparison chart of means and their 95% confidence intervals for ATP concentrations (nmol/g) in cardiac biopsies 1 h after weaning from CPB in the “CPB” and “CPB + NO” groups; (**B**) a comparison chart of means and their 95% confidence intervals for ATP concentrations (nmol/g) in cardiac biopsies 1 h after weaning from CPB in the “CPB + CA” and “CB + CA + NO” groups; (**C**) a comparison chart of means and their 95% confidence intervals for ATP concentrations (nmol/g) in lung biopsies 1 h after weaning from CPB in the “CPB” and “CPB + NO” groups; (**D**) a comparison chart of means and their 95% confidence intervals for ATP concentrations (nmol/g) in lung biopsies 1 h after weaning from CPB in the “CPB + CA” and “CPB + CA + NO” groups.

**Figure 4 biomedicines-12-00719-f004:**
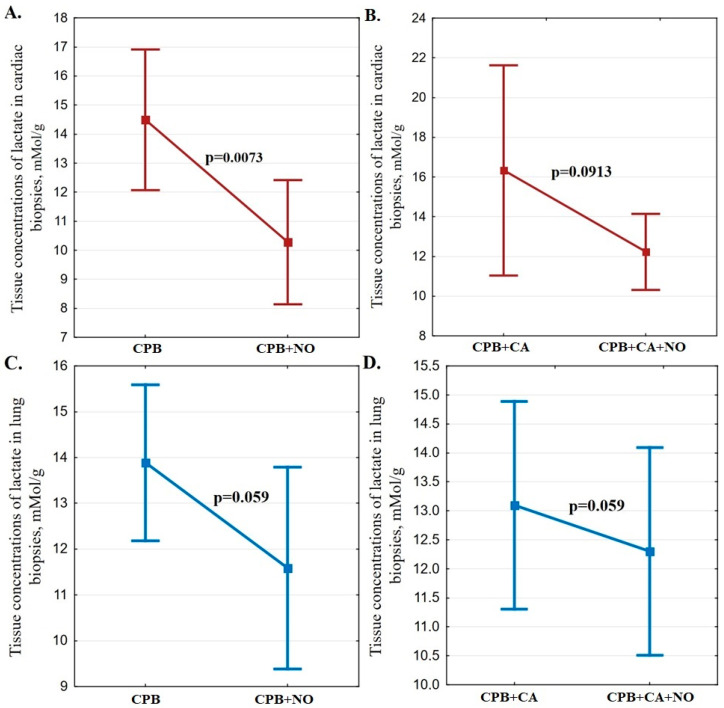
(**A**) A comparison chart of means and their 95% confidence intervals for lactate concentrations (mMol/g) in cardiac biopsies 1 h after weaning from CPB in the “CPB” and “CPB + NO” groups; (**B**) a comparison chart of means and their 95% confidence intervals for lactate concentrations (mMol/g) in cardiac biopsies 1 h after weaning from CPB in the “CPB + CA” and “CPB + CA + NO” groups; (**C**) a comparison chart of means and their 95% confidence intervals for lactate concentrations (mMol/g) in lung biopsies 1 h after weaning from CPB in the “CPB” and “CPB + NO” groups; (**D**) a comparison chart of means and their 95% confidence intervals for lactate concentrations (mMol/g) in lung biopsies1 h after weaning from CPB in the “CPB + CA” and “CPB + CA + NO” groups.

**Table 1 biomedicines-12-00719-t001:** Coefficient of the microviscosity and polarity of erythrocyte membranes before CPB and after weaning from CPB in the “CPB”, “CPB + NO”, “CPB + CA”, and “CPB + CA + NO” groups.

Coefficient of Microviscosity and Polarity of Erythrocyte Membranes in the “CPB” and “CPB + NO” Groups						
Variables	Stages	CPB Group, M ± SD	*p*	CPB + NO Group, M ± SD	*p*	*p* (CPB and CPB + NO Groups)
CMLLI J_470_/J_370_, λ_B_ = 340 nm	before CPB	0.29 ± 0.109	*p* = 0.741	0.26 ± 0.119	*p* = 0.885	*p* = 0.668
	after CPB	0.28 ± 0.101		0.25 ± 0.026		*p* = 0.520
CMPLI J_470_/J_370_, λ_B_ = 285 nm	before CPB	0.40 ± 0.111	*p* = 0.009	0.41 ± 0.165	*p* = 0.827	*p* = 0.946
	after CPB	0.18 ± 0.054		0.39 ± 0.147		*p* = 0.008
CPLLI J_390_/J_370_, λ_B_ = 340 nm	before CPB	1.05 ± 0.011	*p* = 0.616	1.04 ± 0.033	*p* = 0.735	*p* = 0.474
	after CPB	1.05 ± 0.015		1.05 ± 0.017		*p* = 0.869
CPPLI J_390_/J_370_, λ_B_ = 285 nm	before CPB	5.11 ± 0.903	*p* = 0.013	5.14 ± 0.669	*p* = 0.773	*p* = 0.948
	after CPB	3.47 ± 0.534		5.02 ± 0.457		*p* = 0.0003
**Coefficient of Microviscosity and Polarity of Erythrocyte Membranes in the “CPB + CA” and “CPB + CA + NO” Groups**						
**Variables**	**Stages**	**CPB + CA Group,** **M ± SD**	** *p* **	**CPB + CA + NO Group,** **M ± SD**	** *p* **	***p* CPB + CA − CPB + CA + NO Groups**
CMLLI J_470_/J_370_, λ_B_ = 340 nm	before CPB	0.29 ± 0.106	*p* = 0.946	0.30 ± 0.086	*p* = 0.451	*p* = 0.821
	after CPB	0.28 ± 0.061		0.28 ± 0.430		*p* = 0.799
CMPLI J_470_/J_370_, λ_B_ = 285 nm	before CPB	0.40 ± 0.096	*p* = 0.049	0.39 ± 0,079	*p* = 0.946	*p* = 0.847
	after CPB	0.26 ± 0.072		0.39 ± 0.092		*p* = 0.019
CPLLI J_390_/J_370_, λ_B_ = 340 nm	before CPB	1.05 ± 0.033	*p* = 0.882	1.05 ± 0.031	*p* = 0.864	*p* = 0.954
	after CPB	1.05 ± 0.015		1.05 ± 0.015		*p* = 0.727
CPPLI J_390_/J_370_, λ_B_ = 285 nm	before CPB	4.62 ± 1.074	*p* = 0.022	4.51 ± 0.926	*p* = 0.624	*p* = 0.846
	after CPB	2.86 ± 0.559		4.6 ± 0.835		*p* = 0.002

NB: CMLLI—coefficient of microviscosity in the areas of lipid–lipid interactions; CMPLI—coefficient of microviscosity in the areas of protein–lipid interactions; CPLLI—coefficient of polarity in the areas of lipid–lipid interactions; CPPLI—coefficient of polarity in the areas of protein–lipid interactions.

**Table 2 biomedicines-12-00719-t002:** HSPG at different stages of the experiment: before CPB, at the initiation of CPB, and after weaning from CPB in the “CPB”, “CPB + NO”, “CPB + CA”, and “CPB + CA + NO” groups.

HSPG at Different Stages of the Experiment in the “CPB” and “CPB + NO” Groups				
Variable	Stages	“CPB” Group, Me [25; 75]	“CPB + NO” Group, Me [25; 75]	*p*
HSPG, pg/mL	before CPB	22.9 [20.0; 24.1]	21 [20.2; 22.6]	*p* = 0.69
	CPB initiation	22.5 [20.2; 24.8]	23.45 [21.0; 26.6]	*p* = 0.39
	after CPB	25.1 [23.2; 28.7]	23.7 [20.8; 28]	*p* = 0.59
		*p* (before CPB − CPB initiation) = 0.6 *p* (before CPB − after CPB) = 0.17	*p* (before CPB − CPB initiation) = 0.12 *p* (before CPB − after CPB) = 0.35	
**HSPG at Different Stages of the Experiment in the “CPB + CA” and “CPB + CA + NO” Groups**				
**Variable**	**Stages**	**“CPB + CA” Group,** **Me [25; 75]**	**“CPB + CA + NO” Group,** **Me [25; 75]**	** *p* **
HSPG, pg/mL	before CPB	22.81 [20.0; 26.9]	23.4 [20.6; 26.3]	*p* = 1.0
	CPB initiation	20 [18.69; 24.8]	24.4 [22.6; 26.4]	*p* = 0.3
	after CPB	24.2 [21.4; 25.6]	21 [20.7; 23.4]	*p* = 0.39
		*p* (before CPB − CPB initiation) = 0.75 *p* (before CPB − after CPB) = 0.46	*p* (before CPB − CPB initiation) = 0.35 *p* (before CPB − after CPB) = 0.46	

**Table 3 biomedicines-12-00719-t003:** ADMA levels at different stages of the experiment: before CPB, at the initiation of CPB, and after weaning from CPB in the “CPB”, “CPB + NO”, “CPB + CA”, and CPB + CA + NO” groups.

ADMA Levels at Different Stages of the Experiment in the “CPB” and “CPB + NO” Groups				
Variable	Stages	“CPB” Group, M ± SD	“CPB + NO” Group, M ± SD	*p*
ADMA, mkmol/L	before CPB	1.514 ± 0.145	1.575 ± 0.137	*p* = 0.473
	CPB initiation	1.578 ± 1.480	1.584 ± 0.123	*p* = 0.924
	after CPB	1.612 ± 0.123	1.573 ± 0.109	*p* = 0.567
		*p* (before CPB and CPB initiation) = 0.39 *p* (before CPB − after CPB) = 0.24	*p* (before CPB and CPB initiation) = 0.90 *p* (before CPB − after CPB) = 0.98	
**ADMA Levels at Different Stages of the Experiment in the “CPB + CA” and CPB + CA + NO” Groups**				
**Variable**	**Stages**	**“CPB + CA” Group,** **M ± SD**	**“CPB + CA + NO” Group,** **M ± SD**	** *p* **
ADMA, mkmol/L	before CPB	1.580 ± 0.148	1.520 ± 0.120	*p* = 0.461
	CPB initiation	1.636 ± 0.149	1.609 ± 0.0974	*p* = 0.702
	after CPB	1.699 ± 0.154	1.644 ± 0.072	*p* = 0.446
		*p* (before CPB − CPB initiation) = 0.53 *p* (before CPB − after CPB) = 0.20	*p* (before CPB − CPB initiation) = 0.15 *p* (before CPB − after CPB) = 0.06	

**Table 4 biomedicines-12-00719-t004:** ATP and lactate concentrations in cardiac and lung biopsies 1 h after weaning from CPB in the “CPB”, “CPB + NO”, “CPB + CA”, and “CPB + CA + NO” groups.

Variable	Organs	“CPB” Group M ± SD	“CPB + NO” Group M ± SD	*p* “CPB” − “CPB + NO” Groups)	“CPB + CA” Group M ± SD	“CPB + CA + NO” Group M ± SD	*p* “CPB + CA” − “CPB + CA + NO”
ATP, nmol/g	Heart	3.638 ± 0.663	5.983 ± 1.213	*p* = 0.0019	2.875 ± 0.389	3.298 ± 0.575	*p* = 0.1665
	Lungs	3.695 ± 0.725	5.398 ± 0.518	*p* = 0.0008	3.621 ± 0.770	3.263 ± 0.691	*p* = 0.4161
Lactate mMol/g	Heart	14.49 ± 2.31	10.28 ± 2.04	*p* = 0.0073	16.33 ± 5.05	12.24 ± 1.83	*p* = 0.0913
	Lungs	13.89 ± 1.63	11.59 ± 2.10	*p* = 0.059	13.10 ± 1.70	12.30 ± 1.71	*p* = 0.437

## Data Availability

This study includes experimental data, which are available from the corresponding author upon reasonable request.

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
