# Peer review of "Potential Mechanisms for Organoprotective Effects of Exogenous Nitric Oxide in an Experimental Study"

_biomedicines, 2024, doi:10.3390/biomedicines12040719_

Round 1

Reviewer 1 Report

Comments and Suggestions for Authors

Dear Authors, I was reviewing with interest the maniscript entitled "Potential mechanisms for organoprotective effects of exogenous nitric oxide in an experimental study ". You deal with a relevant issue and you use a valid animal model to answer the raised questions. However the question of erythrocyte deformability in combination with lactate does not reflect the actions of NO in the context of CA at all. What about the effect of NO on thrombocytes, which play a major role in this context. I also miss a statement about animal care or ethics approval. In total the study deals with a relevant problem and uses a very interesting intervention (NO), but the presented results are not complete and sound.

Reviewer 2 Report

Comments and Suggestions for Authors

The animal study proved effect of NO in CPB and even CA underwent CPB animals. The study shed lights on avoiding the potential detrimental effect of CPB. The study design was good, and conclusion was firm. I believed the study merit publication. 

Reviewer 3 Report

Comments and Suggestions for Authors

The topic of this manuscript is interesting; however, several deficits need to be addressed as listed below:

1. The scientific writing needs to be carefully checked. There are many errors in English grammar, reference citations, and paragraph arrangement. Please carefully check them.

2. The style of the reference list should be carefully revised.

3. More biochemistry data are required to support the conclusion of this study.

4. This study is more descriptive without molecular mechanistic insights. The authors, at least, should discuss this viewpoint in the revised manuscript.

Comments on the Quality of English Language

Minor editing of English language required

Round 2

Reviewer 1 Report

Comments and Suggestions for Authors

.

Reviewer 3 Report

Comments and Suggestions for Authors

The authors have addressed the reviewer's comments.